# The Amorphous Carbon Thin Films Synthesized by Gas Injection Magnetron Sputtering Technique in Various Gas Atmospheres

**Rafal Chodun [1],\*, Lukasz Skowronski [2], Marek Trzcinski [2], Katarzyna Nowakowska-Langier [3], Krzysztof Kulikowski [1], Mieczyslaw Naparty [2], Michal Radziszewski [1] and Krzysztof Zdunek [1]**

[1] Faculty of Materials Science and Engineering, Warsaw University of Technology, Woloska 141 St., 02-507 Warsaw, Poland

[2] Institute of Mathematics and Physics, Bydgoszcz University of Science and Technology, Al. Prof. S. Kaliskiego 7 St., 85-796 Bydgoszcz, Poland

[3] National Centre for Nuclear Research, Andrzeja Soltana 7, 05-400 Otwock-Swierk, Poland

\* Correspondence: rafal.chodun@pw.edu.pl

**Abstract:** This work presents the potential for using pulsed gas injection to produce amorphous carbon films. In this experiment, the frequency of injecting small amounts of gas was used to control the pressure amplitudes, thus achieving the conditions of plasma generation from stationary, through quasi-stationary, to pulsed oscillations of pressure. In addition, we used various gases and their mixtures, an alternative to argon. In the experiment, we studied the energy state of the plasma. The films were examined for phase and chemical composition, surface morphology, and optical and mechanical properties. We determined low-frequency pulsed gas injections to be conditions favorable for $C(sp^3) - C(sp^3)$ bond formation. The plasma generated by gas injections is better ionized than that generated by static pressure. Pulsed conditions favor the plasma species to retain their kinetic energy, limiting the probability of intermolecular collision events. Since helium has a relatively high ionization energy, it is a practical addition to sputtering gas because of the increasing $sp^3$ content in the films. The electrons created by helium ionization improve the plasma's ionization degree.

**Keywords:** amorphous carbon; a-C; diamond-like carbon; DLC; magnetron sputtering; gas injection; GIMS; Raman spectroscopy; optical properties; nanohardness



## 1. Introduction

Amorphous carbon (a-C) coatings attract the community's attention with exciting properties. Among them are high hardness, good wear resistance, low friction coefficient, transparency, wide band gap, chemical inertness, and biocompatibility [1–3]. Thanks to these properties, amorphous carbon layers are of great interest from the application side: tools, photovoltaics, electronics, photonics, biomedical engineering, etc. [4–10]. The a-C films consist of a mixture of carbon phases with $sp^2$ and $sp^3$ hybridization. The $sp^3/sp^2$ fraction depends on the deposition conditions and determines the material properties of the coating [11,12]. The range of a-C properties is thus extensive, from coatings with graphite-like properties to diamond-like ones. This is an additional advantage of these coatings, as it gives the technological possibility to produce layers precisely with specific properties.

Forming a metastable $C(sp^3) - C(sp^3)$ bond configuration in carbon is difficult due to the high nucleation barrier of diamonds [13]. It is conventionally achieved by applying heat and pressure work. Unconventional approaches use, among other means, the work of an electric field [14]. Among the technological ways of performing unconventional synthesis are the methods of Plasma Surface Engineering [15,16].

Magnetron sputtering (MS) is one of the most popular PVD methods for layer fabrication. Its advantages are versatility, high flexibility in the types of fabricated coatings, chemical purity, homogeneity, and scalability for industrial production [17,18]. Magnetron sputtering, in its conventional direct current MS (dcMS) variant, is characterized by a relatively low degree of ionization of only a few percent, with plasma particle energies reaching only a few eV [19,20]. This makes the a-C layers produced by dcMS have graphite-like features [21,22]. MS technology is being intensively improved, and one of the leading development paths is to take advantage of the benefits of highly ionized plasmas. One variety of MS using high-energy plasma is High Power Impulse Magnetron Sputtering (HiPIMS). In this technique, the plasma is excited in a high-current discharge, its degree of ionization reaches several tens of percent, and the plasma particles reach energies of 100 eV [23–25]. As a result, the a-C layers are characterized by an effective content of $C(sp^3) - C(sp^3)$ bonds and, thus, diamond-like features [26–30].

Another way to increase plasma particle energy is represented by the Gas Injection Magnetron Sputtering (GIMS) technique. The GIMS technique is a pulsed technique that generates discrete plasma streams (plasma pulses) in an environment of dynamically varying working gas concentration, controlled by high-speed pulsed gas valves [31,32]. Typically, in GIMS, layers are produced due to the generation of $10–10^2$ ms plasma pulses at frequencies of $10^{-1}–10^1$ Hz. The pulse power density is of the order of $10^2$ W/cm$^2$. Despite the relatively high power density, the pulsed operation makes the condition for the film's growth at room temperature [33].

GIMS is a practical implementation of the idea of increasing the energy of plasmoids, not as a result of using high-power sources (HiPIMS), but by carrying out the process of deposition under conditions limiting energy dissipation. This is performed by the pulsed method of distributing small portions of gas and using efficient vacuum pumps. Each generated pulse of plasma spreads in a volume of relatively low pressure compared to the MS process at constant pressure. Such conditions favor the preservation of high energies of plasma particles. So far, we have been able to confirm the effectiveness of the GIMS technique:

(1)  By recording the optical emission spectroscopy OES spectra showing high-energy plasma components in GIMS [31,32];
(2)  By recording the effects of the interaction of the energetic GIMS plasma with the substrate surface [31];
(3)  By depositing phases on substrates whose nucleation is characteristic of high-energy phase equilibrium conditions [34–37];
(4)  By depositing non-stoichiometric phases on substrates [38,39].

The advantages of GIMS led it to be implemented in the industry in producing coatings on large-scale, 3 m × 2 m, glass [40,41]. The layers produced on glass using the GIMS technique exhibited better adhesion and uniformity than the standard magnetron sputtering version.

This work aimed to study the GIMS technique's effects in fabricating a-C coatings. We believed that this technique could be particularly effective in obtaining an increased $sp^3$ phase content in the a-C layers. The performance characteristics of the GIMS technique substantiate this assumption. As part of our experiment, we took advantage of the ability to electronically control the operating parameters of the pulsed valves to produce various coating fabrication conditions. The a-C coatings were produced under dynamically fluctuating pressure conditions and quasi-stationary conditions. We also used non-standard gases and their mixtures in our experiment.

Our research procedure characterized the a-C layer synthesis environment and the coatings from structural, phase, and chemical perspectives. In order to obtain a complementary characterization, we studied the properties of the produced layers.

## 2. Materials and Methods

### 2.1. Apparatus

The a-C coatings synthesis was carried out in a vacuum apparatus presented in Figure 1. The vacuum chamber had a vacuum pump system consisting of turbomolecular, roots, and rotary pumps. The vacuum pump unit could reach a $5 \times 10^{-3}$ Pa base pressure. According to the Gencoa criterion [42], a highly unbalanced circular magnetron was used as the plasma source. The 50 mm diameter pyrolytic graphite targets were installed in the magnetron. Figure 1a shows a diagram of an apparatus for synthesizing layers in an argon atmosphere dosed by a single pulse valve into the glow discharge zone above the surface of a graphite target. The operation of the pulse gas valve GV is controlled by a controller coupled to the power supply (PS) of the magnetron gun. The first part of our experiment involved changing the gas dosing frequency parameter. We changed this parameter to the 0.2–1 Hz range in our experiment. In this way, we created conditions for more to less dynamic pressure changes, periodically repetitive with the operation of the pulse valve. Considering the pumping system's limited efficiency, the relatively higher gas dosing frequency (1 Hz) variant created quasi-stationary pressure conditions. The variant of a relatively low gas metering frequency (0.2 Hz) generated highly fluctuating pressure conditions. The qualitative difference between these variants is presented in Figure 1b. The pressure characteristics presented are illustrative. A precise determination of the oscillation was impossible due to the high time inertia of the used vacuum gauge. Pawlak [43] studied a more accurate study of the pressure characteristic in GIMS.

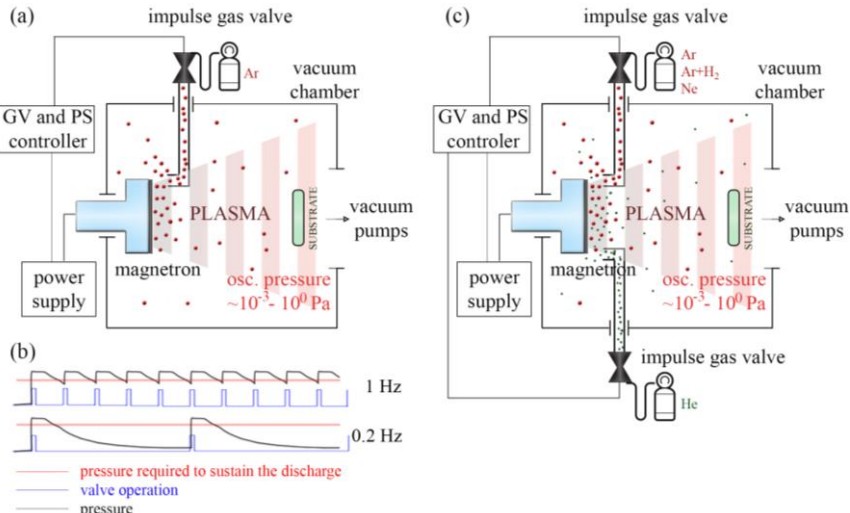

**Figure 1.** Diagram illustrating: (**a**) the apparatus configuration for a-C films deposition in Ar atmosphere controlled by a single pulse valve, (**b**) oscillations depending on the frequency of impulse valve operation, (**c**) the apparatus configuration for a-C films deposition in various gas atmospheres controlled by two pulsed valves.

The second part of the experiment investigated the effects of performing deposition processes in different gaseous atmospheres. For this purpose, two independently controlled pulse valves were used—see Figure 1c.

### 2.2. Coatings Synthesis Processes

The coating deposition processes were carried out using Ar, Ar + $H_2$ (5% $H_2$), Ne, and He gases (N5.0 purity). The working atmosphere was created with injections from a fast pulse valve directly at the target surface. The pressure at the valve inlet was set to $1.5 \times 10^5$ Pa. In the first part of the experiment, the opening time of the valve was fixed at 1 ms to maintain as low-pressure conditions as possible. Ar gas was distributed to the glow discharge region of the magnetron gun with various frequencies, 0.2–1 Hz.

Additionally, a referencing a-C film was fabricated in the conditions of 0.4 Pa constant pressure by the Pulsed Magnetron Sputtering (PMS) technique. This choice is based on the fact that in our GIMS process, the power is distributed as in the PMS mode. The only difference is using a pulse valve that injects gas in a controlled manner. This allowed us to achieve pulsed pressure conditions. The frequency of injections was chosen experimentally. A gas dosing frequency of 0.2 Hz determined conditions where the interval between gas injections was high enough for the pumping system to evacuate gas to a base pressure of $5 \times 10^{-3}$ Pa. A threshold frequency of 1 Hz determined the quasi-stationary conditions, in which the pressure slightly oscillated around a value of 0.4 Pa. For complete characterization, processes were also carried out at transitional frequencies. The deposition process parameters are summarized in Table 1.

**Table 1.** Parameters of a-C coatings deposition in Ar atmosphere.

| Process | PMS | GIMS 1 Hz | GIMS 0.5 Hz | GIMS 0.(3) Hz | GIMS 0.25 Hz | GIMS 0.2 Hz |
|---|---|---|---|---|---|---|
| Valve frequency (Hz) | - | 1 | 0.5 | 0.(3) | 0.25 | 0.2 |
| Time of valve opening (ms) | - | | | 1 | | |
| Pressure (Pa) | 0.4 | | | - | | |
| Pulse of power duration (ms) | - | | | 100 | | |
| Deposition time (min) | 3 | 30 | 60 | 90 | 120 | 150 |

The second part of the experiment examined the effects of using various gases and their mixtures in the synthesizing a-C layers. The a-C layers were fabricated in Ar, Ne, and Ar+$H_2$ atmospheres. In addition, He was used as a component of the base gas mixtures. For the pressure conditions to be reproducible, the sum of the opening times of a single valve or two valves was 2 ms. The process parameters are summarized in Table 2.

**Table 2.** Parameters of a-C coatings deposition in various gas atmospheres.

| Gas 1 | Ar | | Ar + H$_2$ | | Ne | |
|---|---|---|---|---|---|---|
| Gas 2 | - | He | - | He | - | He |
| Time of valve opening (ms) | 2 (Ar) | 1 (Ar) 1 (He) | 2 (Ar + H$_2$) | 2 (Ar + H$_2$) 1 (He) | 2 (Ne) | 2 (Ne) 1 (He) |
| Valve frequency (Hz) | | | 0.2 | | | |
| Pulse of power duration (ms) | | | 100 | | | |
| Deposition time (min) | | | 150 | | | |

The n-type 100 silicon substrates were prepared by ultrasonic cleaning in acetone and drying. After that, they were located perpendicularly to the magnetron Z-axis at 10 cm from its surface. The substrate stage was electrically grounded during coating deposition with the magnetron anode and vacuum chamber. The magnetron system was powered using a Dora Power Systems pulsed DC power supply with a carrier frequency of 125 kHz [44,45].

### 2.3. Plasma Characterization

The current and voltage waveforms of the magnetron power circuit were collected to characterize the discharge. The power waveforms were calculated from the current and voltage characteristics, measured using the 1 A: 0.1 V Rogowski coil and 1 kV: 1 V voltage, and registered by a Rigol oscilloscope. The pulse energy was calculated using the integral over time. All the calculations were made using the Origin Lab software.

The OES measurements were obtained using an Optel energy-dispersive optical spectrometer at a 200–900 nm wavelength. The signals were collected through a quartz window and an optical fiber. The exposition time was set to a full single plasma pulse. During these experiments, the optical collimator was perpendicular to the magnetron Z-axis and placed 55 mm from the cathode surface.

*2.4. Coatings Characterization*

The Raman scattering of the a-C films was studied using a 532 nm VIS laser (2.33 eV) and a 266 nm UV laser (4.66 eV) as the source of VIS and UV excitation, an Ar+ laser and the Crylas FQCW266-50 diode-pumped continuous wave solid-state laser was used, respectively. The scattered light was dispersed by a JASCO NRS 5100 spectrometer working in backscattering mode. The laser beams were focused onto ~15 μm spots during the measurements. The parameters of the spectra registration were optimized based on the excellent peak intensity. No visible damage to the surface of the samples was observed after the Raman examination. In order to analyze the registered spectra, they were processed using the JASCO peak processing software. During the processing, the spectra were interpolated and their background was subtracted to realign the intensity levels at each side of the presented range. The curve fitting procedure included the determination of the peaks *D* and *G* position and a full width at half maximum FWHM. The final intensities, positions, FWHM, and the shapes (Gaussian, Lorenztian, or a mixed Gaussian—Lorentzian function) of any elementary components were optimized by curve fitting software using a least-squares method. The $C(sp^3) - C(sp^3)$ bond content was evaluated based on the *G* peak parameters: *G* peak FWHM and *G* peak dispersion rate [46]. The *sp³/sp²* ratio as the *G* peak dispersion is expressed by:

$$sp^3 content = -0.07 + 2.5 \cdot Disp(G), \tag{1}$$

where:

$$Disp(G) = \left| \frac{Pos(G)@\lambda_2 - Pos(G)@\lambda_1}{\lambda_2 - \lambda_1} \right|, \tag{2}$$

$Pos(G)@\lambda_2$ and $Pos(G)@\lambda_1$ are the *G* peak positions at $\lambda_2$ and $\lambda_1$ irradiation, respectively. $\lambda_2$ and $\lambda_1$ are the laser wavelength.

The *sp³/sp²* ratio as the FWHM criterion can be calculated from:

$$sp^3 content = -0.25 + 1.9 \cdot 10^{-2} \cdot W - 3.01 \cdot 10^{-5} \cdot W^2, \tag{3}$$

where:

$$W = FWHM(G) + 0.21 \cdot (514 - \lambda), \tag{4}$$

and $\lambda$ is the laser wavelength.

In order to investigate the respective contribution of the *sp²* and *sp³* hybridization bonds in the near-surface layers of the samples, an X-ray Photoelectron Spectroscopy (XPS) analysis was performed. Measurements were made under ultra-high vacuum conditions (base pressure ≤ 2·10⁻¹⁰ mbar). The excitation radiation source was an Al K$\alpha$ lamp (1486.6 eV), and the angle of incidence of the excitation beam was 55 degrees. Photoelectron energy was recorded using the VG-Scienta R3000 analyzer. The energy step was set to ΔE = 100 meV. Experimental data were fitted using CasaXPS software using the Shirley background and Gauss-Lorentz line shapes. The C1's level was deconvoluted into four components, the main two related to the $C(sp^2) - C(sp^2)$ and $C(sp^3) - C(sp^3)$ bonds, and two other peaks at higher binding energies from the hydroxyl (C–O) and carboxylic (C=O) bonds from surface impurities.

The AFM measurements were performed using the Innova device from Bruker. The tapping mode was used to investigate the surface topography of the fabricated carbon films. The scan size was 1 μm x 1 μm. The analysis was performed using the Gwyddion software.

The nanohardness measurements were carried out using Vantage Alpha Nanotester (Micromaterials Ltd., UK) equipment. A maximum load of 0.5 mN, loading time of 10 s, dwell period of 5 s, and 10 s unloading time were applied. For every sample, 25 indentations were performed, and the mean values, with standard deviation, were calculated.

The ellipsometric azimuths Ψ and Δ have been recorded in the spectral range from 0.6 eV (2000 nm) to 6.5 eV (190 nm) for three angles of incidence 65°, 70°, and 75°. The

measurements were performed using the V-VASE device from J.A.Woollam Co., Inc. (Lincoln, NE, USA).

## 3. Results and Discussion

Figure 2 shows the power pulse characteristics of the GIMS processes and the characteristics of the MS process under constant pressure conditions. The power pulses were integrated to calculate their energies. In each case, the integration constant was 100 ms. The plot shows that the conditions of the low frequency of gas injection favor plasma generation with higher energy. This is especially evident in the initial stage of the discharge. In our opinion, the amount of energy released during the discharge is influenced by (1) the pressure and (2) the size of the injected gas portion. The favorable conditions for obtaining and sustaining a relatively highly ionized discharge are low pressure in the chamber and a relatively high portion of injected gas. Low chamber pressure limits the dissipation of kinetic energy during collisions of energetic plasma species. The size of the gas portion represents a population of potential particles that can be ionized. The best conditions for a high-energy discharge are those during gas injection.

In our experimental setup, the valve was located outside the vacuum chamber. The gas was delivered via a tube with a diameter of 1 mm and a length of about 200 mm. This makes the gas injection inertially executed. The discharge delay to the start of the valve was 12 ms and, at its beginning, a high amount of gas enters the discharge zone, the concentration of which decreases with time. At the same time, the pressure in the chamber increases as a result of filling it with injected gas. By this, we observe a decrease in the energy of the discharge during the pulse.

Nevertheless, a strong current discharge can be observed at the beginning of the pulse generated at lower pressure conditions. The discharge has a higher energy when lower pressure conditions are achieved in the chamber. Such conditions occur when the process is carried out at a low frequency of gas injection—see Figure 1b. This difference can be seen when comparing the PMS stationary or quasi-stationary GIMS 1 Hz process with the GIMS 0.2 Hz process.

OES studies also showed a qualitative difference between the plasma generation processes under PMS stationary and the GIMS pulsed conditions. Figure 3 shows the recorded OES spectra in the visible VIS and UV ranges. The spectral ranges of excited Ar I particles, Ar II ions, and sputtered C I excited carbon particles were selected. Due to the relatively low resolution of the spectrometer and the slight differences in the lines intensity of all the GIMS processes, it is not possible to make a quantitative assessment of the population size of the individual excited plasma species. Nevertheless, there is a qualitative difference between the intensity of the OES spectrum of the PMS process and the GIMS processes, which may be related to the higher degree of ionization of the plasma produced by the GIMS. The presence of Ar II ions in the OES spectrum of the GIMS processes can also demonstrate this. It seems that the OES results are reflected in the energy state of the plasma pulses determined from the power characteristics of the magnetron power system. The differences in the plasma state seem significant enough that we would expect them to affect the characteristics of the a-C films fabricated under these conditions.

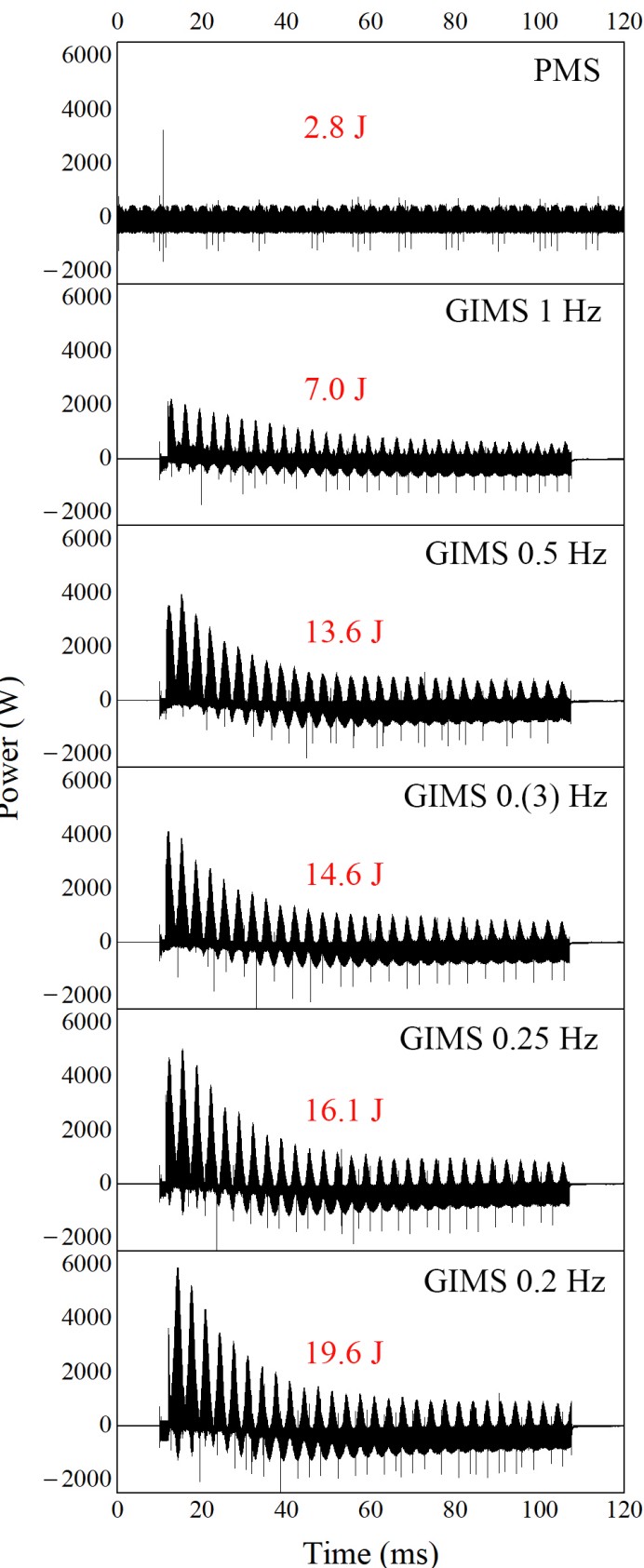

**Figure 2.** The power characteristics measured in the electric circuit of the magnetron gun operating in PMS and various GIMS conditions. The graphs are labeled with the energy values of the entire single power pulses.

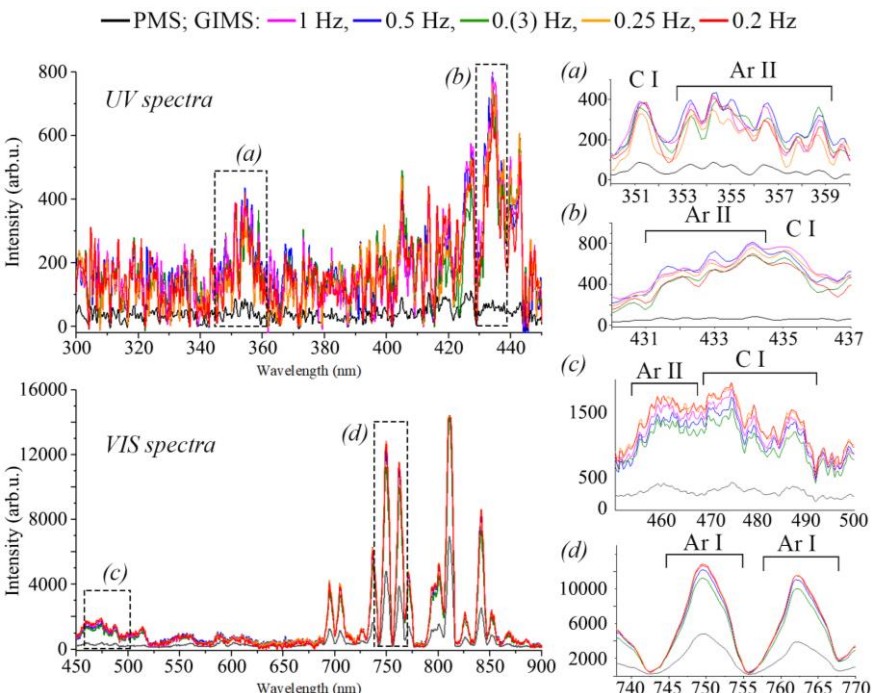

**Figure 3.** UV (**top left**) and VIS (**bottom left**) OES plasma spectra generated in PMS and various GIMS conditions. Selected ranges of the OES spectrum of the characteristic groups of plasma constituents occurrence (**right column**).

Figure 4 shows the Raman spectra of the a-C layers produced under different Ar dosing conditions. The spectra were recorded under VIS and UV laser irradiation. The measured spectra were fitted with two elementary *D* and *G* spectra. The *G* and *D* peaks parameters are shown in Table 3.

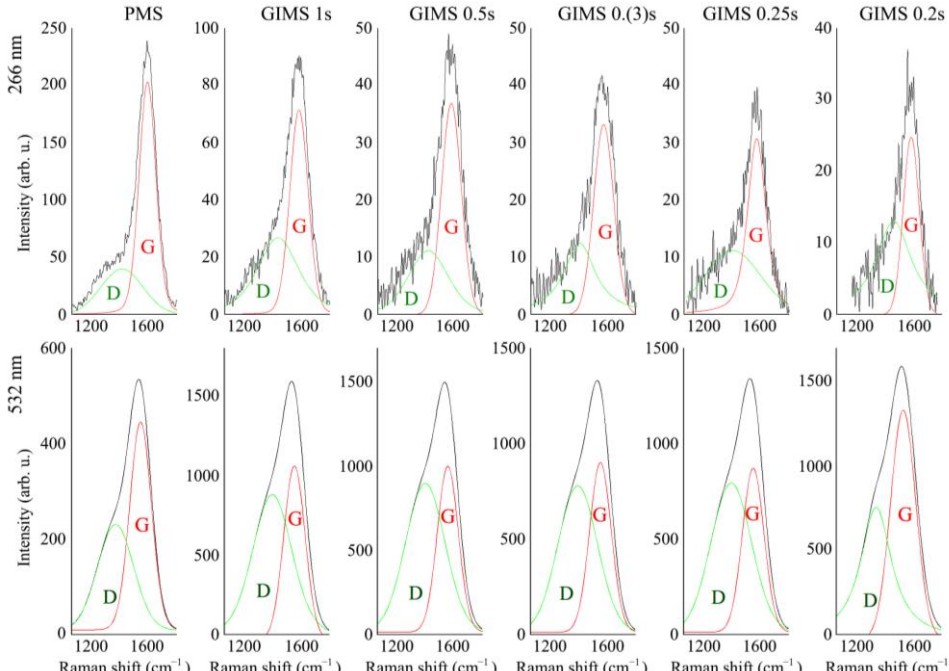

**Figure 4.** Raman spectra of a-C films fabricated in various conditions of Ar dosing, recorded during the irradiation by 266 nm laser (**top**) and 532 nm laser (**bottom**) and fitted with *D* and *G* elementary spectra.

**Table 3.** *G* and *D* peak characteristics of Raman spectra of a-C films deposited in various conditions of Ar dosing.

| Process | Laser | Peak | Position (cm⁻¹) | FWHM (cm⁻¹) | Height (arb. u.) | $sp^3/sp^2$ (G Peak Dispersion Criterion) [44] | $sp^3/sp^2$ (G Peak FWHM Criterion) [44] |
|---|---|---|---|---|---|---|---|
| PMS | 266 nm | G | 1583.88 | 144.25 | 201.98 | | |
| | | D | 1404.25 | 352.11 | 39.91 | 0.19 | 0.21 |
| | 532 nm | G | 1555.44 | 163.40 | 390.91 | | |
| | | D | 1389.43 | 338.13 | 245.68 | | |
| GIMS 1 Hz | 266 nm | G | 1579.84 | 155.31 | 71.33 | | |
| | | D | 1427.54 | 350.66 | 26.85 | 0.20 | 0.35 |
| | 532 nm | G | 1550.16 | 178.70 | 10,562.39 | | |
| | | D | 1391.60 | 338.98 | 8760.61 | | |
| GIMS 0.5 Hz | 266 nm | G | 1574.24 | 166.14 | 36.74 | | |
| | | D | 1407.06 | 337.76 | 11.20 | 0.13 | 0.33 |
| | 532 nm | G | 1552.38 | 176.35 | 9912.99 | | |
| | | D | 1391.07 | 331.55 | 8964.35 | | |
| GIMS 0.(3) Hz | 266 nm | G | 1569.99 | 168.98 | 33.09 | | |
| | | D | 1397.53 | 273.29 | 12.37 | 0.11 | 0.37 |
| | 532 nm | G | 1550.46 | 181.52 | 9615.17 | | |
| | | D | 1387.94 | 331.29 | 8319.19 | | |
| GIMS 0.25 Hz | 266 nm | G | 1569.54 | 155.49 | 30.72 | | |
| | | D | 1390.69 | 473.23 | 11.18 | 0.13 | 0.37 |
| | 532 nm | G | 1548.18 | 181.00 | 9862.05 | | |
| | | D | 1391.01 | 340.68 | 8964.35 | | |
| GIMS 0.2 Hz | 266 nm | G | 1580.48 | 135.29 | 24.71 | | |
| | | D | 1450.91 | 306.81 | 13.13 | 0.42 | 0.46 |
| | 532 nm | G | 1527.50 | 210.35 | 14,067.37 | | |
| | | D | 1333.38 | 268.57 | 7964.90 | | |

Single crystal graphite shows a Raman active band $E_{2g}$, labeled as *G* (Graphite), at ~1580 cm⁻¹. The *G* peak is assigned to the stretching vibration of the $sp^2$ sites. With the graphite amorphization, an extra mode $A_{1g}$ appears at ~1350 cm⁻¹, known as the *D* peak (Disorder). The *D* mode is attributed to the breathing modes of the $sp^2$ sites in rings. The elementary spectrum of peak *D* is slightly shifted towards higher values of cm⁻¹ [47]. We think an additional vibrational mode between peaks *D* and *G*, which was not considered when fitting the spectrum, may be responsible for this. We believe that 1400–1450 cm⁻¹ $C = C$ polyenic vibrations are very likely [48]. According to the Ferrari diagram of 3-stage carbon amorphization [49], the $C(sp^3) - C(sp^3)$ bond is expected when in-plane correlation length $L_s$ is below 2 nm [50]. Correctly determining the characteristics of the *G* and *D* peaks (positions, intensity, and FWHM) from the DLC Raman spectra is reported as the method of evaluating the $C(sp^3) - C(sp^3)$ bond content. The determined peak parameters are relatively similar. Among the results, the 0.2 Hz GIMS process stands out. In this case, a relatively wide FWHM value of the *G* peak and its significant dispersion were determined. The determined $sp^3$ content on this basis reached 42% (dispersion criterion) and 59% (FWHM criterion), respectively. In other cases, the criteria occasionally did not exceed 20% of the $sp^3$ content.

Raman spectroscopy is a standard method for studying carbon materials because it is susceptible [1]. Nonetheless, it is common practice to supplement this with the XPS method.

In order to confirm the results of the Raman spectroscopy, we examined two a-C layers produced in far conditions, PMS and GIMS 0.2 Hz. The XPS results are shown in Figure 5. Only carbon and oxygen were found in the XPS spectra of all the samples. The material contained no nitrogen or other impurities in detectable concentrations. The C1s peak was slightly broadened towards higher binding energies. Its main component in all samples was a peak attributed to the graphite bonds ($sp^2$) occurring at an energy of about

284.8 eV. The less intense component associated with the *sp³* hybridization was found at ca. 285.5 eV. At higher binding energies, characteristic peaks were also present, usually attributed to C–O bonds (ca. 286.6 eV) and C=O (288.8 eV).

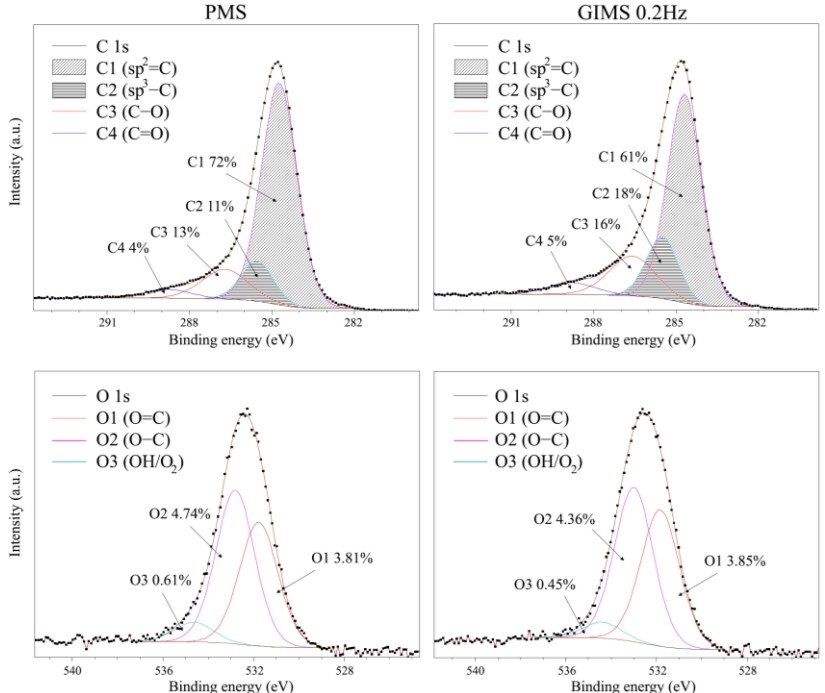

**Figure 5.** XPS spectra of C1s and O1s orbitals of a-C films fabricated at far conditions of gas dosing: stationary PMS (**left**) and pulsed GIMS 0.2 Hz (**right**).

The oxygen *O1s* level was fitted with three components. The most intense was around 533.0 eV, attributed to O–C type bonds. A slightly lower peak from the O=C bonds was found at about 531.8 eV. At 534.4 eV, one more component was perhaps related to water molecules on the surface, OH-type bonds, or oxygen molecules adsorbed on the surface. The detailed composition is presented in Table 4.

**Table 4.** The composition of a-C films determined by X-ray photoelectron spectroscopy.

|  | C1s (%) | | | O1s (%) | | | | Concentration (%) | |
|---|---|---|---|---|---|---|---|---|---|
|  | C ($sp^2$) | C ($sp^3$) | C (C–O) | C (C=O) | O (O–C) | O (O=C) | O (OH) | C | O |
| PMS | 65.20 | 10.75 | 11.45 | 3.44 | 4.74 | 3.81 | 0.61 | 90.84 | 9.16 |
| GIMS 0.2 Hz | 56.50 | 15.72 | 14.72 | 4.40 | 4.36 | 3.85 | 0.45 | 91.34 | 8.66 |

Figure 6 summarizes the *sp³/sp²* ratio determined by the two methods. The determined *sp³* contents clearly show the difference between the films fabricated under extreme conditions, PMS and GIMS 0.2 Hz. The difference is almost two-fold in the method of determining the *sp³* by Raman (~20% to ~45%) and XPS (~15% to ~30%). Intermediate GIMS 1 Hz–GIMS 0.25 Hz conditions resulted in films with an *sp³* content in the mid-range values. The differences in the values calculated by the two methods are due to their limitations. The Raman analysis comes from greater depth than the XPS analysis. It is a familiar fact that the XPS typically shows an increased *sp²* content at the surface [51] due to the proposed mechanism of the *sp³* formation in the a-C layers [52]. Nevertheless, the most important conclusion from this part of the experiment is that the conditions most favorable for generating the a-C layers, with elevated $C(sp^3) - C(sp^3)$ bond content, are those of the GIMS process, in which the plasma is generated in a volume characterized by low pressure.

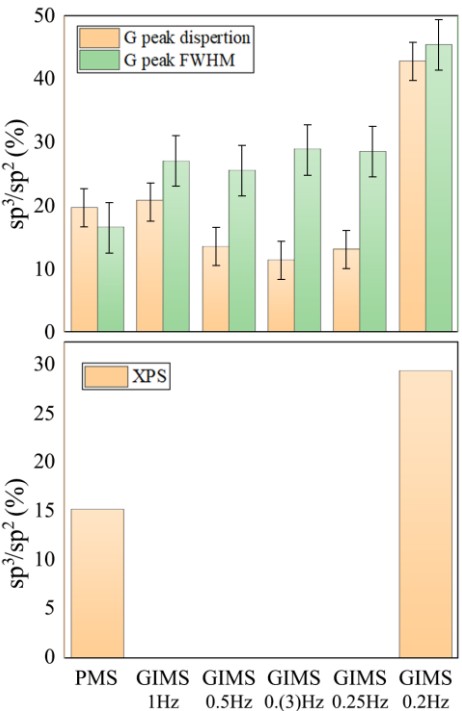

**Figure 6.** Summarized $sp^3/sp^2$ ratio determined by Raman spectroscopy (**top**) and XPS (**bottom**).

We expected that such conditions are particularly favorable for reducing the energy dispersion of plasma particles into inelastic collisions with cold gas molecules. Studies of the state of the plasma and the magnetron power circuit seem to confirm this. Under the conditions of our experimental apparatus, we obtained a good effect with a gas injection frequency of 0.2 Hz. Under these conditions, the highest energy characterized the discharges and we identified relatively high intensities of OES spectra energetic plasma particles in the plasma composition. The formation of an increased fraction of $sp^3$ carbon is usually explained in the literature based on Lifshitz's model of subplantation of hyperthermal species [53,54]. This model explains the formation of a diamond-like phase by impinging the graphitic surface by energetic species with energy ranging from 1 to 100 eV. Energetic species can penetrate the crystal lattice shallowly if their energy is enough. Penetration enables atomic displacements and phonon and electron excitations. Excited lattice carbon atoms get extra energy and the possibility of creating local $sp^3$ hybridization. With an increasing concentration and the kinetic energy of penetrating species, the possibility of forming the $sp^3$ phase increases. This stays behind the observed trend with decreasing the frequency of gas injections. By increasing the energy of plasma, a plasma source generates a larger population of excited species with higher energy, so a stronger impingement of the substrate in this condition is expected.

In the second part of the experiment, we fabricated the a-C layers using various plasma-forming gases injected at a frequency of 0.2 Hz. As in part one, we aimed to figure out what effect a given gas has on the energy of the generated plasma pulses. The results are shown in Figure 7. Using alternatives to Ar gases and their mixtures increases the plasma pulse energy. This is related to the ionization energies of the individual gases.

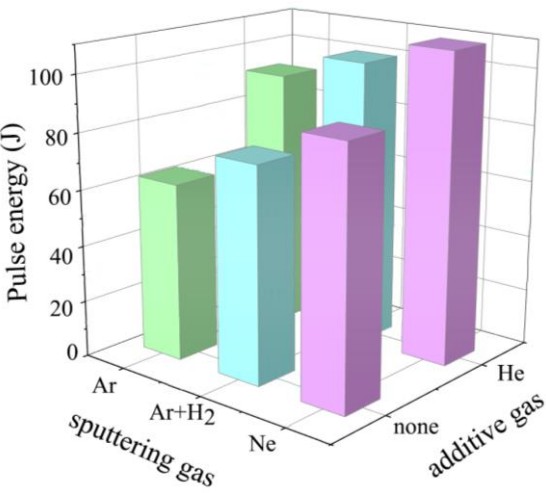

**Figure 7.** The energy of plasma pulse generated in GIMS under various gases and mixture conditions, calculated by integration of the power characteristics.

The ionization energies of Ne ($E_i$ = 21.56 eV) and He ($E_i$ = 24.58 eV) are higher than Ar ($E_i$ = 15.6 eV). Thanks to this, higher energy is characterized by secondary electrons emitted due to ionization. This directly impacts the efficiency of the sputtering and ionization phenomena. The result is an increase in the population of carbon ions. In the last few years, the literature reports using an alternative for Ar gases to increase the ionization degree of magnetron plasma [55–59].

Figure 8 compares AFM images of the surface topography of the a-C films. The layers deposited in the atmosphere, Ar, Ar + H₂, and Ne, are topographically relatively similar. They are dense and smooth. They exhibit topography features characteristic of a-C layers formed by magnetron sputtering. Films obtained in a mixture with He have a different morphology. They are made of globular particles and their structure is characterized by relatively high porosity. Such structures are characteristic of coatings produced at relatively high pressures. Structures built of large agglomerates are formed when a small free path of the particles in the plasma characterizes the process. These conditions favor the agglomeration of the phase into larger forms before reaching the substrate. In the case of our experiment, the reasons for this morphology have to be found somewhere else. All layers were produced under the same gas conditions (injection frequency, valve opening time, valve inlet pressure, etc.). The only difference is the presence of He in the mixture.

In our opinion, the so-called cluster-like growth mechanism is behind obtaining such a globular structure. According to Zdunek, homogeneous phase nucleation occurs directly on the plasma ions in the ion-rich plasma, and thermodynamically stable clusters are formed [14,60]. The clusters can undergo agglomeration to form globular forms delivered to the substrate [61]. Particles on the substrate undergo incomplete coalescence phenomena [60]. Particles that undergo incomplete coalescence can be clearly distinguished in the AFM images. Figure 9 shows a series of images showing different stages of coalescence. Coalescence occurs due to the slight diffusive motion of nanoparticles along the surface, and the mechanism of particle aggregation is through mass transport between particles, activated by plasma energy. Coalescence is not complete due to the pulsed nature of the energy distribution.

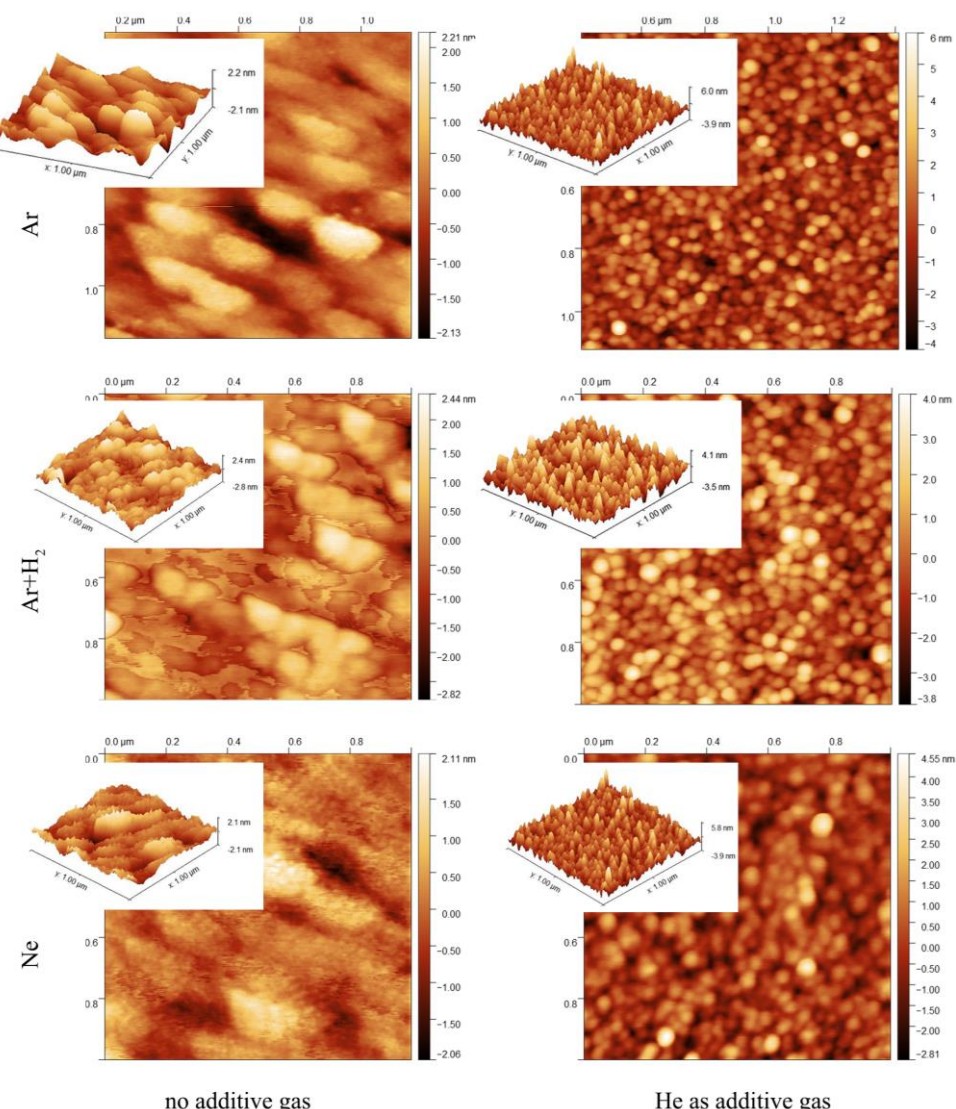

no additive gas                                   He as additive gas

**Figure 8.** AFM images (top view and 3D view) of a-C films topography deposited in various gas atmospheres.

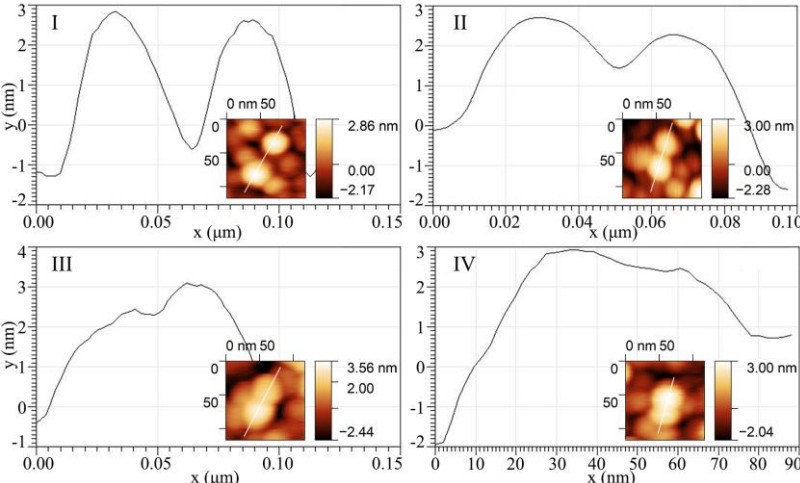

**Figure 9.** Cross-section analysis of carbon nanoparticles AFM images. Sections ordered in I–IV stages presenting the incomplete coalescence of nanoparticles presented as AFM insets.

A prerequisite for homogeneous nucleation to become apparent over heterogeneous nucleation is increased plasma ionization. Investigations of the energy of glow discharge pulses indicate that the conditions of He in the plasma-forming gas mixture contribute to an increase in the discharge energy. The addition of He resulted in an increase in pulse energy in each case. To confirm this, we recorded the OES spectra of plasma excited under these conditions, and the results of this study are presented in Figure 10. The spectra were presented to compare the effect of the presence of He in the plasma gas mixture. In each case, the addition of He contributes to an increase in the intensity of the emission lines. This includes the excited lines of carbon atoms and ions of the plasma-forming gas.

It should be noted that a strong line of CII carbon ions was identified in the plasma generated in a mixture of Ne and He. In all cases, the intensity of the individual emission lines increases by two orders of magnitude.

Of course, it is not possible on this basis to assess how much the population of excited particles in the plasma increases. Nevertheless, it is possible to conclude that the energy state of the plasma excited in gas with He is increased.

Layers formed in different atmospheres of plasma-forming gases were investigated by XPS chemical analysis. The results of the content of the individual bonds and the determined $sp^3/sp^2$ ratios are summarized in Figure 11. The detailed composition is presented in Table 5. First, we could not achieve such $sp^3$ content as in the first part of the experiment.

We believe this is because we had to use two pulse valves to complete the second part of the experiment. With a minimum opening time of 1 ms, the coatings' deposition processes had to be carried out in an atmosphere with twice as much gas, and this caused the negative effect of higher pressure. Although the state of plasma in the individual processes differed significantly, the material effect was not extraordinary. Despite this, the results show a slight increase in the $sp^3$ content in mixtures with He. This corresponds well with the previously reported results, which indicated an elevated plasma energy state in the He environment. It is also worth pointing out that using $H_2$ in the mixture slightly reduced the content of the bonds with oxygen. The presented XPS results show a critical development application for the GIMS technique. There is a clear need for faster pulse valves to produce multi-component gas atmospheres. This is necessary to take advantage of the "genetic" features of GIMS—the advantages of film deposition under low-pressure conditions.

The produced layers were characterized by spectroscopic ellipsometry to characterize the optical properties and to determine the thickness of the produced carbon films. The four medium optical models (from bottom to top: Si\native SiO₂\carbon film\ambient) of a sample were used to obtain the refractive index ($n$) and the extinction coefficient ($k$) of the layers. The optical constants of Si and SiO₂ were taken from the database of optical constants [62]. The thickness of the native silicon dioxide was determined in a separate experiment and was established to be 2.5 nm. The complex refractive index $\tilde{n} = n + ik$ of the carbon film was parameterized using a sum of the Gauss-shaped oscillators:

$$\tilde{n}^2 = \varepsilon_\infty + \sum_j (A_j, E_j, Br_j) \tag{5}$$

where $\varepsilon_\infty$ is the high-frequency dielectric constant, while $A_j$, $E_j$, and $E_j$ are amplitude, energy, and the broadening of the $j$-th oscillator. The fit procedure was performed to minimize the standard mean squared error—MSE [63]. The thicknesses of the fabricated a-C films depended on the mixture of the gas used during the sputtering process and were determined to be 96 ± 1 nm (for Ar), 137 ± 1 nm (for Ar + He), 162 ± 1 nm (for Ar + H₂), 148 ± 1 nm (for Ar + H₂ + Ne), 171 ± 1 nm (for Ne), and 98 ± 1 nm (for Ne + He).

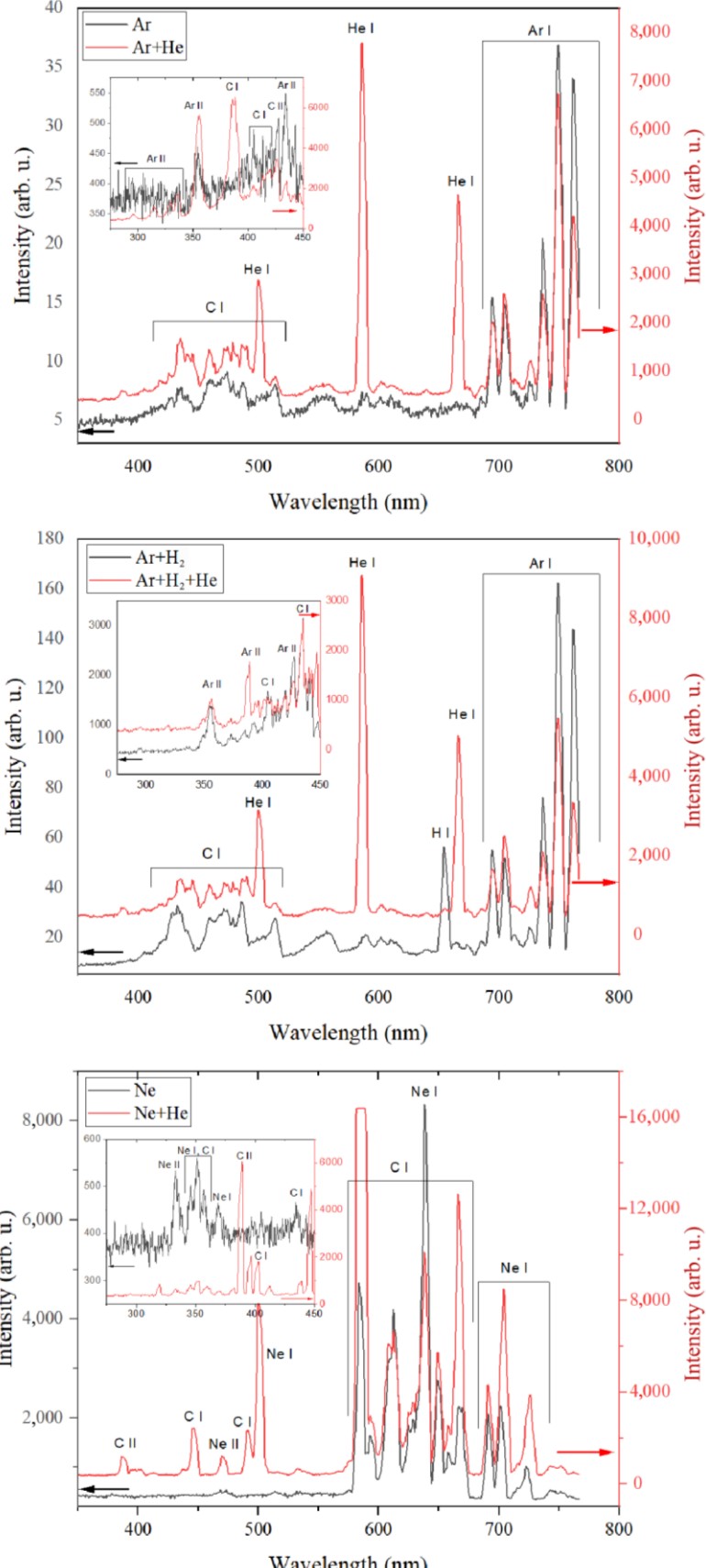

**Figure 10.** OES spectra of plasma pulses generated in various components of gas mixtures. The spectra measured in the VIS range are presented as large figures. Spectra measured in the UV range are presented as smaller insets.

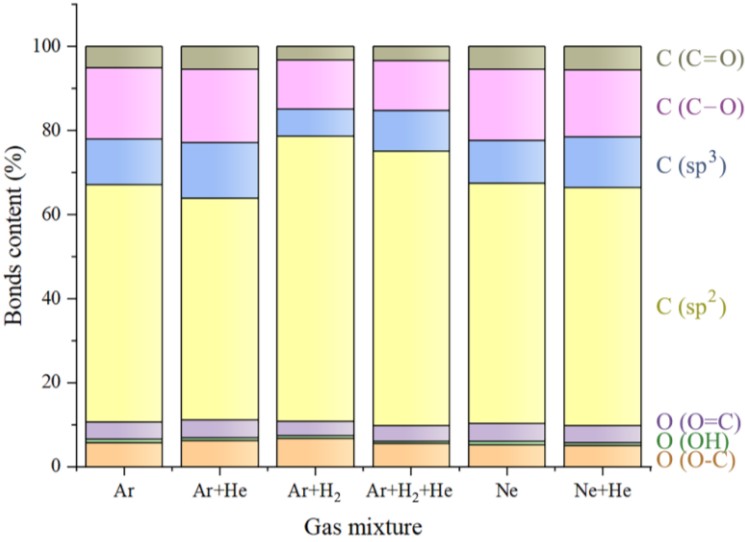

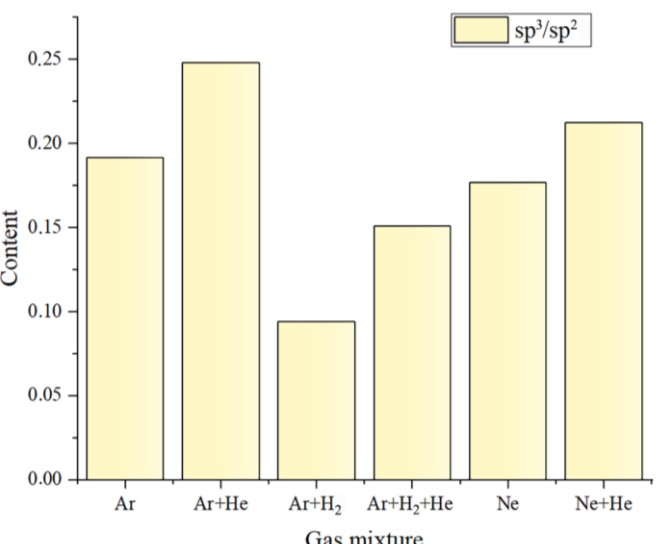

**Figure 11.** Bonding composition of a-C films deposited in various gas atmospheres and determined by XPS (**top**) and calculated $sp^3/sp^2$ ratio (**bottom**).

**Table 5.** The composition of a-C films determined by X-ray photoelectron spectroscopy.

| | C 1s (%) | | | | O 1s (%) | | | Concentration (%) | |
|---|---|---|---|---|---|---|---|---|---|
| | C ($sp^2$) | C ($sp^3$) | C (C–O) | C (C=O) | O (O–C) | O (O=C) | O (OH) | C | O |
| Ar | 57.3 | 10.7 | 16.5 | 5.3 | 5.3 | 4.0 | 0.6 | 89.99 | 10.01 |
| Ar + He | 52.8 | 13.1 | 17.5 | 5.4 | 6.2 | 4.1 | 0.6 | 88.95 | 11.05 |
| Ar + H₂ | 67.8 | 6.3 | 11.7 | 3.2 | 6.7 | 3.3 | 0.6 | 89.21 | 10.79 |
| Ar + H₂ + He | 65.2 | 9.8 | 11.7 | 3.4 | 5.5 | 3.5 | 0.6 | 90.31 | 9.69 |
| Ne | 57.2 | 10.1 | 17.0 | 5.3 | 5.2 | 4.3 | 0.7 | 89.73 | 10.27 |
| Ne + He | 56.7 | 12.0 | 15.8 | 5.6 | 5.0 | 4.0 | 0.7 | 90.22 | 9.78 |

The determined refractive index and extinction coefficient ($n$, $k$) are given in Figure 12. Layers with a higher $C(sp^3) - C(sp^3)$ bond content are characterized by a higher n above ~500 nm, while below ~500 nm, $n$ is lower. In the NIR spectral range, the refractive index values are at 2.5 for the amorphous carbon films with a higher $C(sp^3) - C(sp^3)$ bond content which decreases to about 2 for the film obtained for the Ar + H₂ mixture of sputtered gas. This sample demonstrates the lowest $sp^3$ to $sp^2$ ratio (see Figure 11). The

significant decrease in the *n* spectra for wavelengths shorter than 500 nm is associated with the behavior of the optical response of medium described by the Kramers-Kronig relations [63] of real and imaginary parts of the complex refractive index of the a-C thin films. The absorption features of the deposited carbon films are characterized by the extinction coefficient and absorption spectra ($\alpha$). As can be seen, the highest spectra values are achieved in the UV and gradually decreases with the increase in the wavelength. The *k* reaches higher values for the a-C layers with a higher $C(sp^3) - C(sp^3)$ bond content.

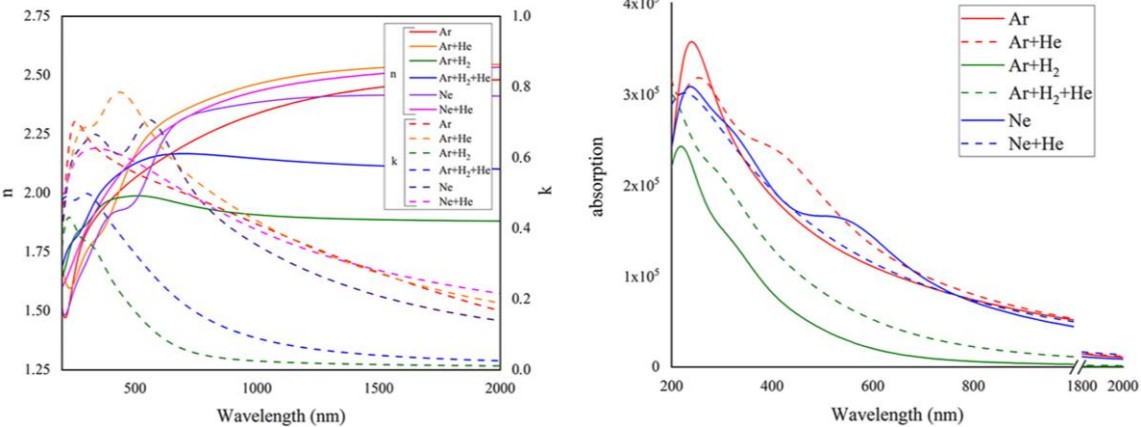

**Figure 12.** Optical characteristics of a-C coatings: (*n*, *k*)—left, absorption—right.

In the UV spectral range, the distinct absorption band ($\pi -> \pi^*$ transition) can be visible in the *k* and $\alpha$ spectra. The lowest amplitude demonstrates the film deposited in the Ar+H₂ mixture (the lowest $sp^3/sp^2$ ratio), and (generally) increases with the increase in the $sp^3$ to $sp^2$ ratio. The TDDFT calculations [62] confirm the behavior of the extinction coefficient/absorption coefficient spectra of the films analyzed in this study. Zoubos et al. show that absorption coefficient values of carbon films with a high (90%) $sp^3$ content decrease rapidly to zero with the increase in the wavelength. The $\alpha$ decrease is less evident for the layers containing a lower $sp^3/sp^2$ ratio, and their values are significantly higher than those for the $sp^3$-rich materials [64]. The deposited a-C layers contain a relatively low content of $sp^3$ compared to the carbon films described in [64]. Therefore, the determined spectra should be treated as the effective optical response of the synthesized a-C films. Moreover, the absorption threshold cannot be precisely determined because of the lack of a pronounced increase in the k or $\alpha$.

The mechanical properties of the a-C films deposited on Si substrates were characterized using nanohardness tests. The results of hardness and reduced Young's modulus measurements using the nanoindentation method are shown in Figure 13. The coatings have a hardness of 8–12 GPa range and a reduced Young's modulus of about 110–160 GPa, which is consistent with the literature results for the a-C layers with a $C(sp^3) - C(sp^3)$ bond content in this range [65–68]. Notably, the summary of the results of the mechanical properties is close to the summary of the $sp^3/sp^2$ ratio shown in Figure 11. This confirms the general knowledge of how much the mechanical properties of carbon films are related to the content of the $C(sp^3) - C(sp^3)$ bonds.

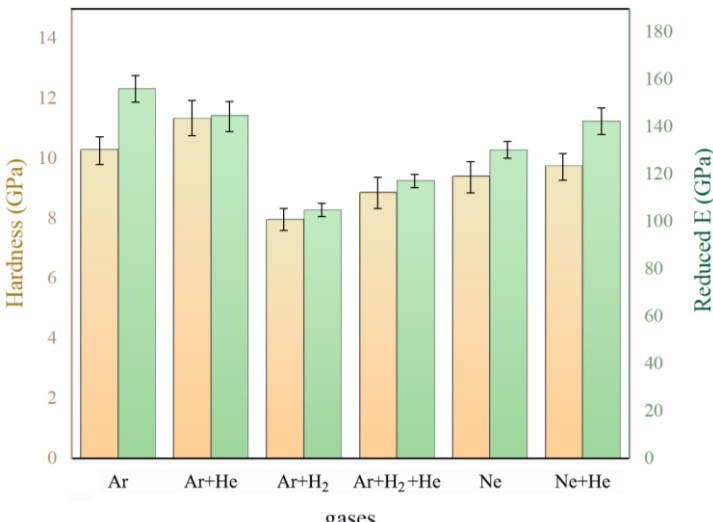

**Figure 13.** Mechanical properties of a-C films expressed by nano hardness and reduced Young modulus.

## 4. Conclusions

This work presented the GIMS technique in the context of the a-C coatings synthesis. The pulsed plasma generated by the injections of a small portion of gas may contribute to preserving the kinetic energy of plasma species. This preservation of the energy is provided when the system, quickly evacuating the cold gas volume, is used. In the GIMS technique, this can be achieved by decreasing the frequency of gas injections. OES investigation proved that plasma generated in this condition consists of more energetic constituents than plasma created in stationary conditions. The plasma emission spectra were compared with the phase structure of the a-C films. In the films fabricated under more energetic plasma conditions, a higher content of the $C(sp^3) - C(sp^3)$ bonds was identified. Using our GIMS experimental setup, we achieved an $sp^3/sp^2$ ratio of ~45%. This is not a result that is achieved by methods such as filtered arc deposition, ion beam deposition, or PECVD methods, where one can approach 90% with the $C(sp^3) - C(sp^3)$ bonds content. We consider our result relatively good, considering that the magnetron sputtering process, in its standard version of power distribution and without substrate polarization, can rarely achieve such values. Only the high-power variant HiPIMS, which additionally uses substrate polarization successfully, exceeds such values. However, it should be noted that the GIMS technique in our experiment used a standard power source. All the results presented in this paper seem consistent and lead to the conclusion that controlling the magnetron plasma, using gas pulses, can be considered a suitable method for optimizing the features of the deposited material in the form of coating.

Moreover, the low–frequency GIMS shows features of a process that uses electric energy more effectively to form the coating material. The second approach used to increase the energy state of the plasma was to use alternative gases. The use of gases with higher ionization energy than argon brings good results. In these cases, we observed discharges with high energies, and the plasma spectrum was enriched with sputtering products and ions. In particular, it seems that He, as an additive to the plasma-forming mixture, is particularly beneficial. Its use resulted in a higher $sp^3/sp^2$ ratio and a slightly different film growth mechanism, characteristic of ion-rich nucleation conditions.

**Author Contributions:** Conceptualization, R.C.; methodology, R.C., K.N.-L., L.S., M.T., M.N., K.K.; software, R.C.; validation, R.C. and L.S.; formal analysis, R.C., L.S., M.T. and K.K.; investigation, R.C., K.N.-L., L.S., M.T., K.K. and M.N.; resources, M.R.; data curation, R.C., L.S., M.T. and K.K.; writing—R.C.; writing—review and editing, R.C.; visualization, R.C.; supervision, R.C. and K.Z.; project administration, K.Z.; funding acquisition, K.Z.. All authors have read and agreed to the published version of the manuscript.

**Funding:** This research was funded by the National Science Centre (grant no. 2018/31/B/ST8/00635).

**Institutional Review Board Statement:** Not applicable.

**Informed Consent Statement:** Not applicable.

**Data Availability Statement:** Not applicable.

**Conflicts of Interest:** The authors declare no conflict of interest.

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
