# Peer review of "The Amorphous Carbon Thin Films Synthesized by Gas Injection Magnetron Sputtering Technique in Various Gas Atmospheres"

_coatings, doi:10.3390/coatings13050827_

Round 1

Reviewer 1 Report

The authors introduced the fabrication of carbon films by using the Magnetron Sputtering technique. The frequency of injecting small amounts of gas was used to control the pressure amplitudes, and the various gases and their mixtures were studied. The authors operated the parameters of pulsed valves to produce various coating fabrication conditions. The films were examined for phase and chemical composition, surface morphology, and optical and mechanical properties.

Before publication, the following issues should be considered:

1.     The authors should write clearly of each figures in the figure captions.

2.     For Raman band, the D band and G band should be compared for optimize sputtering conditions.

3.     More discussions about the Raman spectra should be added.

4.     In figure 5, the assignment of the XPS peak should be discussed in detail.

5.     What’s the right vertical coordinate in Figure 10.

The quality of English is fine.

Author Response

Dear reviewer  

In the beginning, we would like to thank you for your time spent for review our paper. It raised the quality of the submitted article. We have revised the text including your comments. All the amendments were highlighted in yellow. Here is the detailed response to your comments.

“The authors should write clearly of each figures in the figure captions”

We have revised the each figure caption and extended the figures description

“For Raman band, the D band and G band should be compared for optimize sputtering conditions.”

“More discussions about the Raman spectra should be added.”

Thank you for pointing out the lacks in the text related to the presentation of Raman spectroscopy results. We have decided to include tables presenting the most relevant parameters of G and D peaks in the revised version of the article, so that they can be compared with each other. In addition, in the "Experimental" section we added a description of the determination of the sp3/sp2 ratio used in Ref. 44, and we have completed the description to include the changes made. In figure 5, the assignment of the XPS peak should be discussed in detail.

“What’s the right vertical coordinate in Figure 10.”

We have supplied Fig.10 with missing description

Reviewer 2 Report

Overview:

Manuscript (ID: coatings-2350160) “The amorphous carbon thin films synthesized by Gas Injection Magnetron Sputtering technique in various gas atmospheres” reports on the formation of films with variable sp2/sp3 ratio in the conditions of the periodic change of the gas flux. Authors have carried out the characterization of plasma, structural investigations and tribological characteristics. The results are presented in a mostly clear manner; the manuscript is accurately written and well-illustrated. I have some comments which are mostly aimed at the technical issues and the supplementing of several not-so-well-discussed points.

Major comments:

1) Fig.2, GIMS: could you please explain why general decreasing trend is observed (i.e. why 1st gas pulse leads to more efficient energy release than the 10th one) and why the frequency of the power variation is equal for various frequencies of gas injection?

2) You fit the C1s XPS (fig. 5) with C-O and C-C-related lines. However, the elemental composition of the samples is not presented, therefore it is unclear if the samples are oxygenated and/or nitridized, which obscures the correctness of the presented fitting. Is it possible to assess the samples' composition from the survey XPS spectra to make the suggested C1s fitting more grounded?

3) The discussion of the variation of sp3/sp2 content estimated by XPS/Raman spectroscopy presented in lines 257-258 ("The differences in the values calculated by the two methods are due to their limitations, but this seems to be a secondary issue.") is, in my opinion, substandard. XPS is known to downplay the sp3/sp2 ratio of the PVD-deposited a-C coatings, as it is a surface-sensitive method, and the surface of the PVD-deposited a-C films is sp2-rich compared to the volume of the film. This effect is reported in [10.1007/s00339-022-06062-2] (see the discussion related to fig. 6 and Refs.41,47 within). Consider supplementing your discussion taking this effect into account.

4) In fig. 12(b), is it possible to assign the absorption peak at ~320-380 nm to the sp2-carbon-originated π-π* transitions [10.1016/j.solmat.2013.06.019; 10.1016/j.tsf.2021.138966]? Does its intensity correlate with the change of sp2/sp3 ratio? More accurate discussion on the origin of this peak would be interesting for the tailoring of the optical properties of DLC.

Minor comments:

5) In my opinion, “sp3 bonds” is a slang term, as hybridization is a characteristic of atom, not of a bond. Consider the revision: term “C(sp3)-C(sp3) bond” seem more appropriate.

6) In Fig 1, what is "GV and PS"? Additionally, "controller", not "controler".

7) Line 171, define FWHM.

8) "OES" abbreviation should be defined in line 72, not 154.

9) In Fig. 5, Y axis title is missing.

10) In Fig. 10, title of the right Y axes is missing.

11) In Fig. 11 (top subfigure), what is "O1", "O2", and "O3"?

12) Lines 369-372: please discuss how the thickness of the films was estimated.

Author Response

Dear reviewer  

First, we would like to thank you for your time spent reviewing our paper. It raised the quality of the submitted article. We have revised the text, including your comments. All the amendments were highlighted in yellow. Here is the detailed response to your comments.

“Fig.2, GIMS: could you please explain why general decreasing trend is observed (i.e. why 1st gas pulse leads to more efficient energy release than the 10th one) and why the frequency of the power variation is equal for various frequencies of gas injection?”

The pulses that the reviewer pointed out are not related to the gas injection frequency, the carrier frequency of the power supply, or the modulation frequency. The former is 0.2 - 1 Hz, the latter is 125 kHz, and the third is 500 Hz. The visible pulses are related to the power supply's electrical design characteristics. Figure two shows the temporal characteristics of a single discharge sustained due to injecting a single gas portion. The reviewer rightly noted that the initial stage of this discharge releases more power than its end. We briefly explained this in the text. We think two factors influence the amount of energy released in our experiment: 1) the pressure in the chamber and 2) the size of the gas portion.

The favorable conditions for obtaining and sustaining a discharge with a relatively high ionized content are low pressure in the chamber and a relatively high portion of injected gas. Low chamber pressure limits the dissipation of kinetic energy during collisions of energetic particles. The gas portion represents a population of potential particles that can be ionized. The best conditions for a high-energy discharge are those during gas injection. Unfortunately, the valve is outside the chamber in our experimental setup. The gas is delivered via a tube with a diameter of 1mm and a length of about 200mm. This makes gas injection happen with considerable inertia. We measured that the discharge delay to the start of the valve is 12 ms, and at its beginning, a high amount of gas enters the discharge zone, the concentration of which decreases with time. At the same time, the pressure in the chamber increases as a result of filling it with injected gas. By this, we observe a decrease in the energy of the discharge during the pulse. On this basis, it is possible to imagine an ideal system in which the gas is injected in a portion where all components are ionized and do not have to overcome the additional path from the valve. Relative to the experiment presented in the article, we are currently testing pulse valves used for fuel injection in internal combustion engines. They are much faster <1ms; we mount them directly in the chamber on the magnetron construction. This is closer to an ideal solution, and we will soon be able to present it in our next article. In the current one, we have slightly rewritten this section so that its description and interpretation of the results are better understood.

“You fit the C1s XPS (fig. 5) with C-O and C-C-related lines. However, the elemental composition of the samples is not presented, therefore it is unclear if the samples are oxygenated and/or nitridized, which obscures the correctness of the presented fitting. Is it possible to assess the samples' composition from the survey XPS spectra to make the suggested C1s fitting more grounded?”

We agree with these comments. The description should be more accurate and consider the elemental composition. We have completed the description of the XPS results. We have given the composition in Tables 4 and 5. We have modified Fig.5 by adding the O 1s orbital and Fig.11 by describing oxygen bonds.

“The discussion of the variation of sp3/sp2 content estimated by XPS/Raman spectroscopy presented in lines 257-258 ("The differences in the values calculated by the two methods are due to their limitations, but this seems to be a secondary issue.") is, in my opinion, substandard. XPS is known to downplay the sp3/sp2 ratio of the PVD-deposited a-C coatings, as it is a surface-sensitive method, and the surface of the PVD-deposited a-C films is sp2-rich compared to the volume of the film. This effect is reported in [10.1007/s00339-022-06062-2] (see the discussion related to fig. 6 and Refs.41,47 within). Consider supplementing your discussion taking this effect into account.”

We sincerely thank you for the advice. We have used it and expanded the discussion to include the indicated issue

“In fig. 12(b), is it possible to assign the absorption peak at ~320-380 nm to the sp2-carbon-originated π-π* transitions [10.1016/j.solmat.2013.06.019; 10.1016/j.tsf.2021.138966]? Does its intensity correlate with the change of sp2/sp3 ratio? More accurate discussion on the origin of this peak would be interesting for the tailoring of the optical properties of DLC.”

Thank you very much for this valuable comment. We have used it in the discussion.

“In my opinion, “sp3 bonds” is a slang term, as hybridization is a characteristic of atom, not of a bond. Consider the revision: term “C(sp3)-C(sp3) bond” seem more appropriate.”

“In Fig 1, what is "GV and PS"? Additionally, "controller", not "controler".”

“Line 171, define FWHM.”

“"OES" abbreviation should be defined in line 72, not 154.”

“In Fig. 5, Y axis title is missing.”

“In Fig. 10, title of the right Y axes is missing.”

“In Fig. 11 (top subfigure), what is "O1", "O2", and "O3"?”

All the minor suggestion have been considered in the revision, and amendments were made.

“Lines 369-372: please discuss how the thickness of the films was estimated.”

Layer thicknesses were determined by ellipsometry spectroscopy and are given in line 562 - 565 of the revised text

Reviewer 3 Report

The presented manuscript discusses original results on amorphous carbon coatings synthesis using Gas Injection Magnetron Sputtering (GIMS). The manuscript is well-prepared and easy to follow, and the quality of the figures is sufficient for most cases. It was shown that plasma generated by gas injections is better ionized than that generated by static pressure and these conditions were more favorable for sp3 bond formation. It was also experimentally justified that to increase the energy state of the plasma it is worth considering using alternative gases and gas mixtures to argon.

However, to increase the value of the manuscript it would be useful to clarify a few points before publishing:

1.      As GIMS is a practical implementation of the idea of increasing the energy of plasmoids not because of using high-power sources (HiPIMS), why Pulsed Magnetron Sputtering (PMS) technique was used as for referencing in this work (not HiPIMS)?

2.      How a-C coatings received in this work using GIMS (for example in terms of sp3/sp2 ratio) are comparable with the a-C coatings received using other methods (CVD, HiPIMS etc.). Would be useful to add a few sentences on it in the results and discussion part.

3.      What about the industrial application of GIMS? Is it possible to receive coatings on large surfaces? Is the deposition rate comparable with other methods? etc.

Author Response

Dear reviewer  

First, we would like to thank you for your time spent reviewing our paper. It raised the quality of the submitted article. We have revised the text, including your comments. All the amendments were highlighted in yellow. Here is the detailed response to your comments.

“As GIMS is a practical implementation of the idea of increasing the energy of plasmoids not because of using high-power sources (HiPIMS), why Pulsed Magnetron Sputtering (PMS) technique was used as for referencing in this work (not HiPIMS)?”

We chose PMS for two reasons. First, from our experience, GIMS is closer to PMS than to HiPIMS in terms of the degree of ionization. Second, in our GIMS experimental setup, we use the same power source as PMS, so de facto, in our GIMS process, the power is distributed as in the PMS way. The difference is the use of pulsed gas injection, which affects the process. It is worth mentioning that pulsed gas injection can be used in all ways of power distribution. We are currently working on using a gas injection in the HiPIMS process. In the revised text, we have decided to add a brief mention of why we refer GIMS to PMS.

“How a-C coatings received in this work using GIMS (for example, in terms of sp3/sp2 ratio) are comparable with the a-C coatings received using other methods (CVD, HiPIMS etc.). Would be useful to add a few sentences on it in the results and discussion part.”

Thank you for this comment. We agree that an overall comparison of results to other techniques is essential. We decided it was worth adding a brief note about this in the Conclusions section.

“What about the industrial application of GIMS? Is it possible to receive coatings on large surfaces? Is the deposition rate comparable with other methods? etc.”

GIMS has been implemented in the industry in producing large-scale 3x2m window glass coatings. The layers produced on glass using the GIMS technique exhibited better adhesion and uniformity than PMS. We decided to add information about the use of GIMS in the industry in a revised version of the article. The issue of deposition efficiency is relevant in the context of pulsed methods, and undoubtedly GIMS is not as good as the PMS method. Nevertheless, GIMS works at a duty cycle of 10-90%, which is quite a good result.

Round 2

Reviewer 2 Report

Authors have addressed my comments and substantially improved the manuscript. I appreciate authors' rigorous feedback. However, I have some additional minor suggestions regarding the discussed data.

1) Table 3 presents the Raman D-peak positions of 1390-1450 cm-1, and they are significantly shifted from the position range of 1320-1380 cm-1 generally observed for D-lines [10.3390/magnetochemistry8120171, Fig. 11-up in 10.1103/PhysRevB.61.14095]. Could authors comment on that? May some other lines, such as polyenic C=C vibrations (1400-1450 cm-1) [10.1016/j.tsf.2021.138966], contribute to the spectra, shifting the positions to higher wavelengths? Polyene formation may take place due to the plasma-induced decomposition of residual hydrocarbons.

2) Line 362: Raman spectra don't always come from the entire thickness of the film. Raman analysis depth depends on the optical/UV transmission and absorbance of the material. It is better to reformulate the sentence, stating that Raman spectroscopy analysis depth is typically larger than the one of XPS.

3) In the legend of fig. 10, in "H2" notation, 2 should be in the subscript mode.

4) In fig. 11, revise the subscript/superscript mode of "H2", "sp2" and "sp3" terms.

Author Response

Dear reviewer  

We sincerely thank you for yet another thorough review of our paper and for providing new suggestions. Below is the response to the review. We have made all suggested changes to the text and drawings. The changes are marked in yellow.

"Table 3 presents the Raman D-peak positions of 1390-1450 cm-1, and they are significantly shifted from the position range of 1320-1380 cm-1 generally observed for D-lines [10.3390/magnetochemistry8120171, Fig. 11-up in 10.1103/PhysRevB.61.14095]. Could authors comment on that? May some other lines, such as polyenic C=C vibrations (1400-1450 cm-1) [10.1016/j.tsf.2021.138966], contribute to the spectra, shifting the positions to higher wavelengths? Polyene formation may take place due to the plasma-induced decomposition of residual hydrocarbons."

We agree with the reviewer's suggestion about shifting the D peak toward higher values and that this requires commentary in the text. While fitting our spectra, I noticed that there is space between the G and D peaks for additional vibration mods. I am not a big fan of coming to far-fetched conclusions from fitting additional mods in these locations, as I am unsure which ones to include. Most likely, the reviewer is correct, and there are vibrations such as polyenic, but you might as well fit wagging CH modes in there; it is still worth considering binding to oxygen. Another difficulty is to model the right peak between the G and D peaks. When both peaks are close to each other and pretty broad, I am unsure what parameters of this sought-after mode to use. I think that Raman spectroscopy, especially the apparatus I use, and my skills do not allow me to do this fitting in a way that would not raise doubts. In such cases, however, I prefer to focus on obvious things, such as the parameters of the G peak. In addition, I make use of XPS studies. In our article, the ratio of sp3 to sp2 was tracked as a consequence of the influence of the synthesis environment on the film formation mechanism. It was not our goal to characterize additional factors and develop additional threads. We did not want to distract from the main thread. Nevertheless, we take the suggestion seriously and think that the behavior of peak D is worth explaining in the text, which we did.

"Line 362: Raman spectra don't always come from the entire thickness of the film. Raman analysis depth depends on the optical/UV transmission and absorbance of the material. It is better to reformulate the sentence, stating that Raman spectroscopy analysis depth is typically larger than the one of XPS."

Thank you for pointing out that we used an over-generalizing statement regarding the depth of penetration and radiation scattering in Raman spectroscopy. We wrote this because, in the case of our films, the measurement covered the full depth of the layer (we observed peaks of silicon, which was the substrate). We have corrected this sentence as suggested.

"In the legend of Fig. 10, in "H2" notation, 2 should be in the subscript mode."

"In fig. 11, revise the subscript/superscript mode of "H2", "sp2" and "sp3" terms."

Thank you for spotting these errors in the figures. We have included the corrected illustrations in the revised version of the article.